# Antioxidant Capacity and Protective Effect of Cow Placenta Extract on D-Galactose-Induced Skin Aging in Mice

**DOI:** 10.3390/nu14214659

**Published:** 2022-11-03

**Authors:** Liu-Hong Shen, Lei Fan, Yue Zhang, Yu Shen, Zhe-Tong Su, Guang-Neng Peng, Jun-Liang Deng, Zhi-Jun Zhong, Xiao-Feng Wu, Shu-Min Yu, Sui-Zhong Cao, Xiao-Lan Zong

**Affiliations:** 1The Key Laboratory of Animal Disease and Human Health of Sichuan Province, The Medical Research Center for Cow Disease, College of Veterinary Medicine, Sichuan Agricultural University, Chengdu 611130, China; 2College of Pharmacy, Chengdu University of Traditional Chinese Medicine, Chengdu 610075, China

**Keywords:** cow, placenta, D-galactose, antioxidant, skin aging

## Abstract

Placental extract has been used for skin care and delaying skin aging. Cow placenta is an abundant resource with a large mass, which has not been harnessed effectively. Cow placenta extract (CPE) has the functions of antioxidation, anti-inflammatory, promoting growth and development, and promoting hair growth. However, little is known about the effect of oral administration of cow placenta extract on skin conditions. Therefore, the present study aimed to investigate the antioxidant capacity of CPE in vitro and in vivo and its protective effect on d-galactose (D-gal) induced skin aging in mice. The results showed that CPE had strong free radical scavenging, reducing and metal chelating activities. CPE can increase the activity of catalase (CAT), glutathione peroxidase (GSH-Px), peroxidase (POD), superoxide dismutase (SOD), and the content of glutathione (GSH), decrease the content of malondialdehyde (MDA). Moreover, CPE can decrease the gene and protein expression of matrix metalloproteinase 1a (MMP-1a) and matrix metalloproteinase 3 (MMP-3) and increase the expression of transforming growth factor-β (TGF-β) and tissue inhibitor of metalloproteinase 1 (TIMP-1) of mouse skin. Histopathological analysis showed CPE reduced the collagen damage caused by D-gal, increased collagen synthesis and reduced its degradation to delay skin aging.

## 1. Introduction

The skin is the largest organ by surface area in the human body and acts as the primordial barrier of the organism against the outside environment, which exerts multiple key functions, including protection, immune response and neuroendocrine [1]. With increasing age, the skin develops visible signs of aging, presenting skin elasticity deteriorated, hydration weakened, transepidermal water loss increased, and collagen content decreased [2]. Therefore, that knowledge in preventing aging is, in general, of high importance, where finding new products and using already available sources may provide good research in this area. Studies have reported that human placenta and porcine placenta extract have good skin care and antidermal-aging effects [3,4]. Cow placenta contains multiple biologically active substances such as amino acids, trace elements, hormones, and cytokines. At present, we have used liquid chromatograph-mass spectrometer (LC-MS)/mass spectrometer (MS) to analyze the differences in antioxidant activity of cow placenta extract (CPE) by different proteases, and investigated the bioactive polypeptides of dairy cows’ placenta and established an effective enzymatic hydrolysis evaluation index for preparing bioactive hydrolysates [5,6]. These results suggest that CPE has a strong anti-aging potential.

Currently, there are some reports on oral intake of anti-aging drugs. Oral intake of porcine placenta extract can improve human skin quality and increase skin elasticity, hydration and barrier function [3]. Oral intake of chicken bone collagen peptides anti-skin aging in mice by regulating collagen degradation and synthesis, inhibiting inflammation and activating lysosomes [7]. In addition, studies have evidenced that the main bioactive substances of placenta extract are polypeptides, which have favorable antioxidant activity [8,9,10]. Oxidative damage is the main contributor to skin aging processes. Reactive oxygen species (ROS) is the primary cause of oxidative damage and plays an essential role in skin aging. ROS can damage DNA and RNA, resulting in reduced translation efficiency and adenosine triphosphate (ATP) synthesis, which in turn causes abnormal collagen and elastin synthesis, eventually leading to skin aging [11,12]. Therefore, exploring interventions to improve antioxidant capacity can effectively prevent skin aging. D-galactose (D-gal) is a powerful glycosylation agent which can destroy elastin and collagen, the major structural molecules of the skin [13,14], cause oxidative stress, disturb redox homeostasis, and ultimately lead to the induction of severe oxidative damage [14,15]. Therefore, D-gal-treated mice are widely considered an ideal animal model for studying oxidative damage and skin aging.

Therefore, this study determined the free radical scavenging power, reducing power and metal chelating power of CPE by 2,2′-azino-bis (3-ethylbenzothiazoline-6-sulfonic acid; ABTS), ferric reducing antioxidant power (FRAP), and Ferrozine methods. Further, we clarified the antioxidant activity and anti-aging effects of CPE in the D-gal-induced mouse model by detecting serum antioxidant indexes, histopathological skin changes, aging-related genes and protein levels.

## 2. Materials and Methods

### 2.1. Placenta Collection and CPE Preparation

Fresh placenta of normally parturated, healthy Holstein cows, aged 3–5 years, weighing above 600 kg and parity 2–4, was collected from a dairy farm in Sichuan Province, China. Then immediately washed with normal saline to remove residual blood and dirt in the placenta and stored at −20 °C.

Preparation of CPE according to our previous method [6]. The placenta was thawed at room temperature and minced. Then, the placenta was homogenized in deionized water by a homogenizer (FSH 2A homogenizer; Yuexin, China) in an ice bath to obtain placenta homogenate. Next, hydrolyzed placenta homogenate by papain (800 U/mg, Shanghai Yuanye Bio-Technology Co., Ltd., Shanghai, China) at neutral pH and 55 °C with 35.74% substrate concentration and 3.92% enzyme-substrate ratio for 5.49 h. Inactivate papain at 90 °C for 10 min, then centrifuged (5427 R centrifuge, Eppendorf, Hamburg, Germany) the solution at 9200× *g* for 5 min and collected the supernatant. Finally, the supernatant was freeze-dried for 48 h to prepare CPE (LyoQuest-55, Telstar, Terrassa, Barcelona).

### 2.2. Antioxidant Capacity In Vitro

0.1 g CPE was dissolved in 1 mL sterile deionized water and diluted to 10, 20, 30, 40 and 50 mg/mL for further experiments.

#### 2.2.1. ABTS Radical Scavenging Activity

Diammonium 2,2′-azinobis[3-ethyl-2,3-dihydrobenzothiazole-6-sulphonate (ABTS) kit was purchased from Beyotime Biotechnology (S0119, Nantong, China). ABTS radical scavenging activity was assayed according to the manufacturer’s instructions.

#### 2.2.2. Ferric Reducing Antioxidant Power (FRAP)

FRAP kit was purchased from Beyotime Biotechnology (S0116, Nantong, China). FRAP radical scavenging activity was assayed according to the manufacturer’s instructions.

#### 2.2.3. Metal Chelating Ability

The metal chelating ability of CPE was assayed according to the method of Noon et al. [16]. Ferrous sulfate (FeSO_4_) (10 μL 4 mmol/L) was added into 0.5 mL CPE solution and reacted for 3 min at room temperature. Ferrozine (10 μL of 20 mmol/L) was then added and reacted for 10 min in darkness. Finally, absorbance (A) was measured at 562 nm. Results were expressed as the Ferrozine inhibition rate (%).
(1)$$\mathrm{Ferrozine}\;\mathrm{inhibition}\;\mathrm{rate}\;\left(\%\right)=\frac{\mathrm{Acontrol}-\mathrm{Asample}}{\mathrm{Acontrol}}\times 100\%$$

### 2.3. Animals and Treatment

Forty 8-week-old clean-grade Kunming mice were fed under standard conditions for 1 week with food and water available ad libitum, then randomly divided into four groups, 10 mice/group (*n* = 10): negative control group (group NC), model group (group M), treatment group (group CPE) and positive control group (group VC). Group M, CPE and VC were injected intraperitoneally with D-gal, 500 mg/kg/day, and the NC group was injected intraperitoneally with an equal amount of saline. Group CPE was gavaged with CPE, 2000 mg/kg/day; group VC was gavaged with vitamin C (Vit C), 100 mg/kg/day; group NC and M were gavaged with equal amounts of distilled water, respectively. The body weight of mice was measured weekly to adjust the injection dose. After 8 weeks of continuous treatment, they were given 3 days to recover.

Criteria for successful modeling: Mice showed numbness, mental inactivity, unresponsiveness, rough hair, skin thinning, laxity and weight loss; skin sections showed a significant decrease in skin thickness and dermal layer thickness.

### 2.4. Serum and Tissue Preparation

At the end of restorative feeding, the blood samples were collected by enucleation of the mice’s eyes and sacrificed by cervical dislocation immediately. Two samples of 1 cm × 1 cm depilated skin tissue were collected from the back, one sample was fixed in 4 % paraformaldehyde, and the other sample was stored at −80 °C. Blood was placed in a centrifuge tube without anticoagulant, stood for 30 min, then centrifuged at 9200× *g* for 5 min to separate the serum, and the serum was collected and stored at −80 °C.

### 2.5. Oxidation-Associated Biomarkers Determination in Mouse Serum

The activity of catalase (CAT), glutathione peroxidase (GSH-Px), peroxidase (POD), superoxide dismutase (SOD), and the content of glutathione (GSH), malondialdehyde (MDA) in mouse serum were determined by relative kits (purchased from Beijing Solarbio Science & Technology Co., Ltd., Beijing, China) as reference.

### 2.6. HE and Masson Stain of Mouse Skin

Fixed skin tissue in 4% paraformaldehyde for 48 h, then embedded in paraffin. For hematoxylin-eosin (HE) staining, skins were sliced into 5 μm, dehydrated with gradient ethanol and stained with hematoxylin and eosin. Masson staining: stained skins in hematoxylin for 2–5 min, washed with water, differentiated with hydrochloric alcohol, blued with ammonia water, and rinsed with running water; stained in 1% phosphomolybdic acid for 1–2 min, controlled time under the microscope to see red muscle fibers and light red collagen fibers; stained directly into aniline blue staining solution for 2–5 min. Dried in a 60 °C oven, cleared with xylene, and sealed with neutral resin.

### 2.7. MMP-1a, MMP-3, TIMP-1 and TGF-β Protein Expression in Mouse Skin

Added skin tissue and phosphate buffered saline (PBS) in a tissue grinding tube at a ratio of 1:9 and homogenized thoroughly until no visible tissue debris remained. The tube was centrifuged at 4600× *g* for 20 min. Finally, the supernatant was collected and used for further analysis.

The ELISA kits were purchased from Nanjing Jiancheng Bioengineering Institute (Nanjing, China). The level of matrix metalloproteinase 1a (MMP-1a), matrix metalloproteinase 3 (MMP-3), tissue inhibitor of metalloproteinase 1 (TIMP-1), and transforming growth factor-β (TGF-β) in skin tissue were measured according to the manufacturer’s instructions. Briefly, samples and standards were added to the 96-well plates, and biotin antigen was added and reacted at 37 °C for 30 min. The plate was washed 5 times and then incubated with horseradish peroxidase (HRP)-conjugate and reacted at 37 °C for 30 min. The plate was washed 5 times, and Chromogen solutions A and B were added and incubated at 37 °C for 10 min in darkness. Finally, a stop solution was added to each well, and after 10 min, the optical density at 450 nm was measured. The standard curve regression equation was calculated by ELISA Calc software according to the standard concentration and optical density (OD) value, and curve fitting was performed using four-parameter logistic regression.

### 2.8. MMP-1a, MMP-3, TIMP-1 and TGF-β Gene Expression in Mouse Skin

Total RNA was extracted from skin tissue with an animal total RNA isolation kit (Nanjing Jiancheng Bioengineering Institute, Nanjing, China) and quantified by measuring the absorbance at 260 nm. Then, total RNA was processed for reverse transcription using Transcriptor First Strand cDNA Synthesis Kit (Sigma Aldrich, Roche, Germany).

In order to carry out the quantitative expression of mRNA, real-time quantitative polymerase chain reaction (qPCR) amplification was implemented on an FQD-96A Detection system (BIOER, Hangzhou, China) with Stormstar SybrGreen qPCR Master Mix (DBI Bioscience, Ludwigshafen, Germany). The data were analyzed by the 2^−ΔΔCT^ method and normalized to the expression levels of glyceraldehyde-3-phosphate dehydrogenase (GAPDH). The primer sequences used for real-time qPCR are listed in Table 1.

### 2.9. Statistical Analysis Statistical

Statistical analysis of data was performed with the one-way ANOVA analysis using SPSS 26.0 software (IBM SPSS Statistics, San Jose, CA, USA). Differences between groups were analyzed by Duncan’s multiple polar difference test. For all tests, *p* < 0.05 was considered statistically significant; *p* < 0.01 was highly significant. Figures were created with GraphPad Prism version 8.0 for Windows (GraphPad Software, San Diego, CA, USA).

## 3. Results

### 3.1. Antioxidant Activity of CPE In Vitro

We measured the antioxidant capacity of CPE in vitro from three different aspects, including ABTS radical scavenging activity, FRAP, and Metal chelating ability. The Trolox standard curve was: y = 0.8484x + 0.0755, R2 = 0.9924 (Figure 1A). The Trolox-Equivalent antioxidant capacity (TEAC) of CPE increased with increasing concentration, and the average TEAC of CPE was 19.56 μmol/g (Figure 1C). The FeSO_4_ standard curve was: y = 0.6489x + 0.08267, R2 = 0.9914 (Figure 1B). CPE reducing power was positively correlated with concentration. The mean value of the reducing power of CPE was 43.62 μmoL/g (Figure 1D). The inhibition rate of Ferrozine increased with the concentration of CPE, and the increase slowed down when the concentration of CPE reached 30 mg/mL. The inhibition rate of Ferrozine was up to 72.60% in the experiment concentration range (Figure 1E). The above results indicated that CPE had good antioxidant activity in vitro, which provided a theoretical basis for further in vivo research.

### 3.2. Effect of CPE on Body Weight

The rate of weight gain in group M decreased compared to the group NC after the third week, while that of the group CPE was higher than the other three groups after the second week. The weight gain of group VC was similar to group NC (Figure 2A). The final weight of each group is shown in Figure 2B. The weight of group M was lower than the other three groups and was significantly lower than group NC (*p* < 0.05); group CPE had a significant increase in weight compared with group M (*p* < 0.05). The results showed that CPE had a certain growth-promoting effect.

### 3.3. Effect of CPE on Oxidation-Associated Biomarkers in Serum

The activities of CAT, GSH-Px, POD and SOD were significantly lower (*p* < 0.05) or extremely significantly lower (*p* < 0.01) (Figure 3A–D), the level of GSH was extremely significantly lower (*p* < 0.01) (Figure 3E), and the level of MDA was extremely significantly higher (*p* < 0.01) (Figure 3F) in the group M than those in the group NC. Administering CPE or Vit C effectively increased CAT, GSH-Px, POD and SOD activities (*p* < 0.01) (Figure 3A–D), GSH level (*p* < 0.01) (Figure 3E), and decreased MDA level (*p* < 0.01) (Figure 3F), compared to those in the group M. These results indicated that PE had strong antioxidant activity, which was equivalent to a positive control (Vit C).

### 3.4. Effect of CPE on Skin Tissue Construct

Group M had damaged epidermal structures, fewer visible hair follicles, significantly reduced dermal thickness compared to group NC (*p* < 0.05), and underwent typical aging pathological damage (Figure 4A). Dermal thickness was significantly higher, and the number of hair follicles was more in group CPE than in group M (*p* < 0.05). The dermal thickness was higher in group VC compared to group M, but the difference was not significant (*p* > 0.05; Figure 4B,C). Thus, CPE improves the histological structures of the skin by reducing epidermis damage and increasing the number of hair follicles and the thickness of the dermis.

The collagen fibers in group M were mildly lax, collagen fibers became thin, broken and shorter, and the collagen fiber area was significantly lower than that of group NC (*p* > 0.05) (Figure 5A,B). The group CPE and VC were richer in collagen fibers with CPE and Vit C interventions, respectively, and no significant laxity or breakage was seen, and the area of collagen fibers was significantly increased compared with the group M (*p* > 0.05) (Figure 5A,B), indicating that PE improved D-gal-induced pathological changes to restore the normal structure of collagen fibers.

### 3.5. Effect of CPE on MMP-1a, MMP-3, TIMP-1 and TGF-β Gene and Protein Expression in Mice Skin

MMP-1a expression was significantly increased (*p* < 0.05), and MMP-3 was extremely significantly increased (*p* < 0.01) (Figure 6A,B), while TGF-β was extremely significantly decreased (*p* < 0.01) and TIMP-1 expression was decreased in the group M compared with the group NC, but the differences were not significant (*p* > 0.05) (Figure 6C,D). MMP-1a was significantly decreased (*p* < 0.05), MMP-3 was extremely significantly decreased (*p* < 0.01) (Figure 6A,B), TGF-β was extremely significantly increased (*p* < 0.01), and TIMP-1 was significantly increased (*p* < 0.05) (Figure 6C,D) in the group CPE compared with the group M, all of which were not significantly different from the group NC (*p* > 0.05).

MMP-1a and MMP-3 levels were significantly higher (*p* < 0.05) (Figure 7A,B), and TIMP-1 levels were significantly lower (*p* < 0.05) (Figure 7D) in group M compared with group NC. MMP-1a and MMP-3 levels were highly significantly lower (*p* < 0.01), TIMP-1 levels were significantly higher (*p* < 0.05), and TGF-β was elevated in the group CPE and VC compared with the group M, with no significant difference (*p* > 0.05) (Figure 7A–D). Based on the above results, CPE can alleviate skin aging by decreasing the gene and protein expression of MMP-1a and MMP-3 and increasing the gene and protein expression of TGF-β and TIMP-1.

## 4. Discussion

### 4.1. Antioxidant Activity of CPE In Vitro

Studies have confirmed that the placenta has strong antioxidant activity, and its main antioxidant components include uracil, tyrosine, phenylalanine, tryptophan, collagen in human placenta [17,18,19], water-soluble protein in the porcine placenta [10], and polypeptide in goat placenta [8]. The dairy cow placenta has great similarities with other mammals and the human placenta, but there are few reports on its research. We have used LC-MS/Ms technology to qualitatively and quantitatively analyze the enzymatic hydrolysates of dairy cows. The main components are peptides with antioxidant capacity, which contain leucine, lysine, isoleucine and arginine [5]. These determine the potential of CPE as a natural antioxidant. In addition, we also compared the antioxidant activity and Vit C equivalent of different enzymatic hydrolysates and obtained the optimal extraction method, which provided theoretical support for this study [20].

The peptides obtained from goat placenta have a good scavenging effect on 2,2-diphenyl-1-picrylhydrazyl (DPPH) free radicals in a concentration-dependent manner [8]. Porcine placenta hydrolysate had an excellent reducing capacity, metal chelating ability and higher DPPH and ABTS inhibitory activities [21]. The results of this study indicated that the ABTS radical scavenging power of CPE is 19.56 μmol/g, the FRAP reducing the power of CPE is 43.62 μmol/g, and the Ferrozine inhibition rate of 50 mg/mL CPE can reach 72.60%. In conclusion, CPE has good antioxidant activity in vitro. However, the in vitro antioxidant activity of CPE prepared in this study was weaker than that of the above placenta extract. We speculate that CPE was prepared by enzymatic digestion of papain in this study, so it contains a variety of peptides of unequal length and has a large molecular weight. In contrast, most of the placental peptides reported in the data were prepared by purification methods such as ultrafiltration or chromatographic separation and have a small molecular weight. However, peptides with smaller molecular weights have higher antioxidant activity [22,23], so the antioxidant activity of CPE is slightly inferior to other animal placental peptides.

### 4.2. Effect of CPE on the Body Weight of Mice

As age increases, the decreasing physiological functions will finally lead to weight loss [24]. Studies showed that the weight gain in D-gal-induced aging mice was significantly lower than that in healthy mice [25,26]. Our study showed that weight gain in D-gal-induced aging mice began to slow down from week 3 and was significantly lower than that in healthy mice at week 4–8, which was consistent with the above results. The weight gain in Vit *C*-treated mice was similar to that in healthy mice, indicating that Vit C could delay the weight loss caused by D-gal. Moreover, as shown in Figure 2, the body weight gain of mice in CPE-treated was higher than that in Vit *C*-treated mice and healthy mice, and the final weight was significantly higher than that of D-gal-treated mice (*p* < 0.05). Bioactive peptides play an important nutritional role, have a positive impact on changes in body composition and muscular performance, and can reduce muscle damage following exercise [27]. Our study showed that CPE could improve the weight loss induced by D-gal, exert beneficial health effects, and has significant potential as a nutritional supplement.

### 4.3. Effect of CPE on Serum Antioxidant Indexes and in Mice

MDA is the product of membrane peroxidation and is considered an indicator of lipid peroxidation and membrane damage [27]. In contrast, GSH, GSH-Px, POD and SOD protect host cells from oxidative damage by scavenging free radicals in vivo [28]. Research data have demonstrated that there was a significant decrease in GSH, SOD and CAT in serum, heart, liver, lungs, kidney and brain tissues during D-gal-induced aging, a significant increase in MDA and an increase in oxidative damage in the organism [25,29,30]. The results of this study demonstrated that serum CAT, GSH-Px, POD, SOD and GSH were significantly decreased (*p* < 0.05) or extremely significantly decreased (*p* < 0.01), and MDA was highly significantly increased (*p* < 0.01) in D-gal-treated mice compared to healthy mice, indicating that D-gal had destroyed the redox homeostasis and was consistent with the above findings.

Carnosine and taurine treatments decreased MDA and protein carbonyl levels and elevated GSH levels, SOD and GSH-Px activities ameliorated histopathological findings in the livers of D-gal-treated rats, which exerts protective actions [31]. The administration of fibroblast growth factor or l-Theanine significantly alleviated histological lesions induced by D-gal and suppressed profound elevation of ROS production and oxidative stress by reducing MDA and GSH levels and restoring SOD, CAT and GSH-Px activities in the liver [32,33]. Tilapia skin collagen polypeptide could alleviate d-gal-induced histopathological impairments to the liver and kidneys. Moreover, it improved the activities of SOD, CAT and GSH-Px, and inhibited the increases in MDA [26]. Our study showed that CPE and Vit C significantly increased CAT, GSH-Px, POD, SOD activities and GSH levels (*p* < 0.01) and significantly decreased MDA levels (*p* < 0.01) compared to D-gal-induced model mice. These findings suggested that CPE has a similar effect to Vit C in regulating the redox balance of D-gal-induced aging mice by restoring antioxidant enzyme activity and reducing peroxidation.

### 4.4. Effect of CPE on Skin Aging-Related Markers in Mice

With aging, skin collagen and elastin will perform characteristic changes that indicate skin aging, such as collagen fiber thinning, crosslinking, breaking, elastin deposition etc. In addition, decreased collagen and elastin synthesis can lead to decreased skin elasticity and increased wrinkles, showing typical skin aging symptoms [13]. Therefore, collagen and elastin are important indicators for studying skin aging. This study showed that the epidermal structure of D-gal-treated mice was damaged, and the number of hair follicles decreased. The thickness of the dermis and the area of collagen fibers in D-gal-treated mice were significantly lower than those in healthy mice (*p* < 0.05), which was consistent with the results of Jiming Chen and Zejun Zhang, indicating that D-gal damaged the skin structure of mice and caused pathological changes similar to skin aging [34,35]. Oral intake of porcine placenta extract can improve human skin quality and increase skin elasticity, hydration and barrier functions [3]. Oral intake of chicken bone collagen peptides can improve the condition of aging skin in mice, including maintaining a smooth, orderly, and complete structure and increasing the dermal thickness and collagen content [7]. Our results showed that the skin structure of CPE mice was similar to that in healthy mice, and no obvious pathological damage was found. The thickness of the dermis and the area of collagen fibers in CPE mice were significantly higher than those in D-gal model mice.

MMP-1 and MMP-3 are interstitial collagenases that degrade type I collagen and type III collagen, which are the most abundant collagen in the skin [36]. In addition, MMP-3 can extensively degrade extracellular matrix and activate other MMPs [37]. The increase of MMP-1 and MMP-3 leads to excessive degradation of collagen, which is an essential factor leading to skin aging. TIMP is an endogenous inhibitor of MMPs, so it is a critical regulator of the extracellular matrix. TIMP-1 is an inhibitor of MMP-1, can promote the growth of keratinocytes and fibroblasts, and plays a vital role in skin aging [38]. TGF-β is a potent growth factor that controls the expression, deposition and turnover of collagen and other extracellular matrix proteins in the skin [36], which can induce paracrine and autocrine signal pathways at transcriptional and translational levels, as well as the expression of other growth factors. The inhibition of the pathway controlled by TGF-β leads to a decrease in collagen synthesis. Our study showed that MMP-1a and MMP-3 expression were increased, TGF-β and TIMP-1 were decreased in D-gal treated mice compared with healthy mice, which indicated that D-gal could promote skin aging by increasing the level of MMPs, decreasing the level of TIMP-1 and TGF-β, and regulating the synthesis and degradation of collagen, which is consistent with the results of Andres and Zhang Xie [39,40]. Conversely, compared with D-gal treated mice, the gene and protein expression of MMP-1a and MMP-3 were significantly decreased (*p* < 0.05) or extremely significantly decreased (*p* < 0.01), the expression of the TGF-β gene was significantly increased (*p* < 0.01), and the content was increased, but not significantly (*p* > 0.05) in CPE treated mice. In addition, the expression and content of TIMP-1 were significantly increased (*p* < 0.05). The above results were similar to those of Vit *C*-treated mice. In summary, CPE can promote collagen synthesis and reduce collagen degradation, thereby alleviating skin damage caused by D-gal, and the effect is equivalent to Vit C.

## 5. Conclusions

In this present study, we evaluate the antioxidant activity of CPE in vivo and in vitro and its anti-aging activity based on the mouse skin aging model induced by D-gal. In vitro, CPE has a strong free radical scavenging ability, reducing power and metal chelating ability. In vivo, it can enhance the antioxidant enzyme activities, including CAT, GSH-Px, POD and SOD in mouse serum, increase GSH level, and reduce MDA content. Furthermore, it can improve the histological damage of skin induced by D-gal, reduce the expression of MMP-1 and MMP-3, increase the expression of TGF-β and TIMP-1, and then regulate the production and degradation of collagen to delay skin aging.

This study provides a theoretical basis for implementing cow placenta extract to alleviate skin aging and broaden the scope for the comprehensive utilization of animal by-products in functional foods. To the best of our knowledge, this is the first study to clinically evaluate the effect of oral intake of cow placenta extract on skin conditions. This study has a few limitations. We need to detect more skin health-related biomarkers and further study the mechanism of CPE delaying skin aging to comprehensively evaluate the beneficial effects of CPE, but we can speculate that CPE alleviates skin aging by regulating antioxidant signaling pathways and collagen production pathways. For prospects, we will further study the molecular mechanism of CPE anti-aging and toxicity in vitro and in vivo, including antioxidant, anti-inflammatory and anti-apoptosis.

## Figures and Tables

**Figure 1 nutrients-14-04659-f001:**
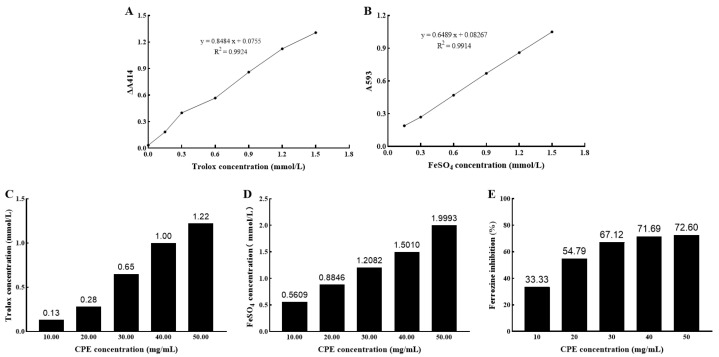
Antioxidant capacity of CPE in vitro. Note: (**A**): Trolox standard curve; (**B**): FeSO_4_ standard curve; (**C**): TEAC of 10–50 mg/mL CPE; (**D**): FeSO_4_ concentration corresponding to 10–50 mg/mL CPE; (**E**): Metal chelating capacity of 10–50 mg/mL CPE. CPE, cow placenta extract; A, absorbance; ∆, change; FeSO_4_, ferrous sulfate; TEAC, Trolox-Equivalent antioxidant capacity.

**Figure 2 nutrients-14-04659-f002:**
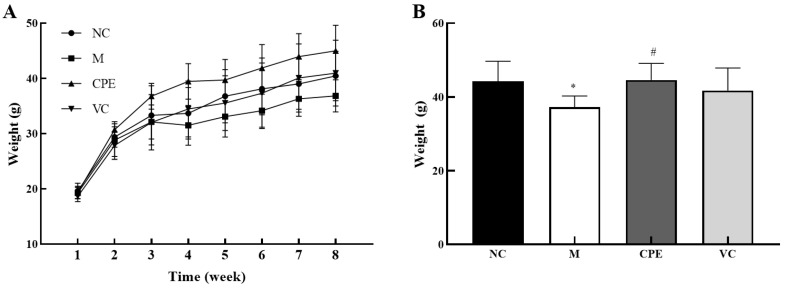
Body weight changes of mice during the experiment. Note: (**A**): the weight growth of mice during the experiment; (**B**): the weight of mice at the eighth week. “*” indicated that the date had statistically significant differences (*p* < 0.05) compared with group NC. “#” indicated that the date had statistically significant differences (*p* < 0.05) compared with group M. NC, negative control group; M, model group; CPE, treatment group; VC, positive control group.

**Figure 3 nutrients-14-04659-f003:**
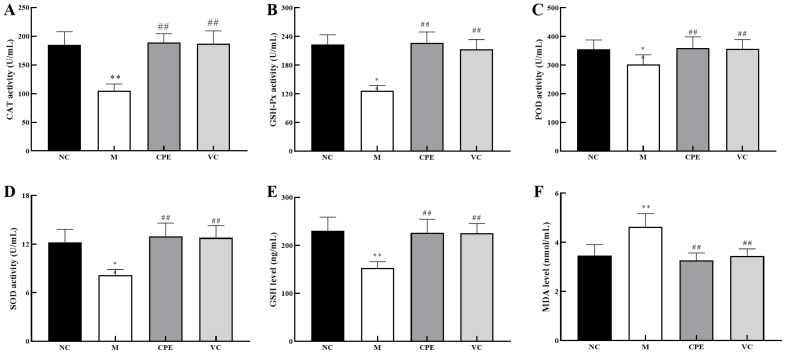
Antioxidant capacity of CPE in the serum. Note: (**A**): Serum CAT activity in each group; (**B**): Serum GSH-Px level in each group; (**C**): Serum POD activity in each group; (**D**): Serum SOD level in each group; (**E**): Serum GSH activity in each group; (**F**): Serum MDA activity in each group. “*” and “**” indicated that the date had statistically significant differences (*p* < 0.05) and extremely significant (*p* < 0.01) compared with group NC, respectively. “##” indicated that the date had extremely significant (*p* < 0.01) compared with group M. CAT, catalase; GSH-Px, glutathione peroxidase; POD, peroxidase; SOD, superoxide dismutase; GSH, glutathione; MDA, malondialdehyde.

**Figure 4 nutrients-14-04659-f004:**
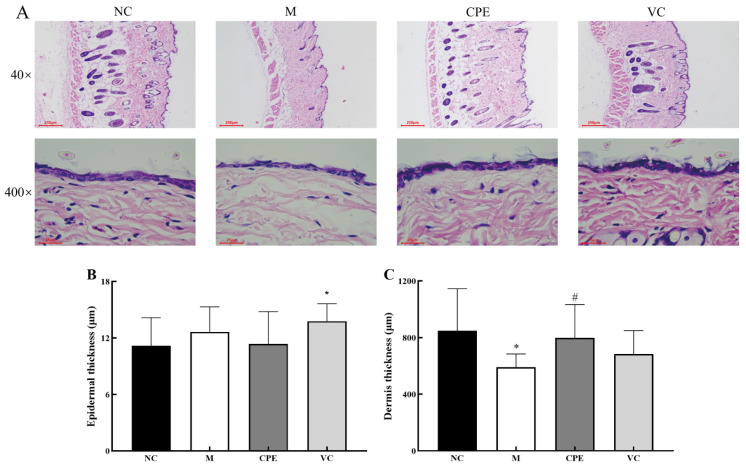
Results of epidermal and dermal thickness in mice skin with HE staining. Note: (**A**): HE-stained sections of skin tissue in each group; (**B**): Epidermal thickness of skin tissue in each group; (**C**): Dermal thickness of skin tissue in each group. “*” indicated that the date had statistically significant differences (*p* < 0.05) compared with group NC. “#” indicated that the date had statistically significant differences (*p* < 0.05) compared with group M. HE, hematoxylin-eosin.

**Figure 5 nutrients-14-04659-f005:**
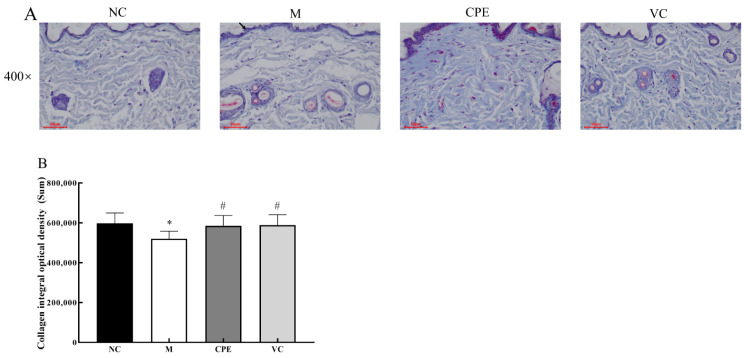
Results of collagen in mice skin with Masson staining. Note: (**A**): Masson -stained sections of skin tissue in each group; (**B**): Collagen IOD of skin tissue in each group. “*” indicated that the date had statistically significant differences (*p* < 0.05) compared with group NC. “#” indicated that the date had statistically significant differences (*p* < 0.05) compared with group M. IOD, integral optical density.

**Figure 6 nutrients-14-04659-f006:**
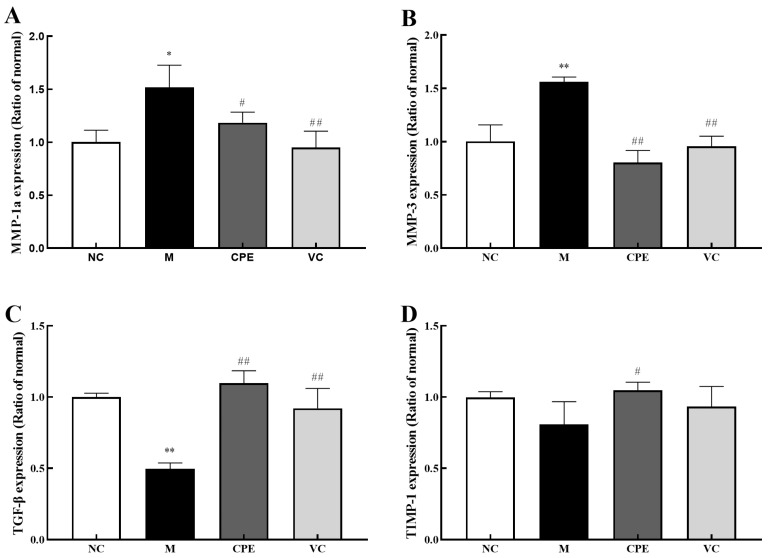
MMP-1a, MMP-3, TIMP-1 and TGF-β levels in mice skin. Note: (**A**): MMP-1a content in mouse skin tissue; (**B**): MMP-3 content in mouse skin tissue; (**C**): TGF-β content in mouse skin tissue; (**D**): TIMP-1 content in mouse skin tissue. “*” and “**” indicated that the date had statistically significant differences (*p* < 0.05) and extremely significant (*p* < 0.01) compared with group NC, respectively. “#” and “##” indicated that the date had statistically significant differences (*p* < 0.05) and extremely significant (*p* < 0.01) compared with group M, respectively. MMP-1a, matrix metalloproteinase 1a; MMP-3, matrix metalloproteinase 3; TGF-β, transforming growth factor-β; TIMP-1, metalloproteinase 1.

**Figure 7 nutrients-14-04659-f007:**
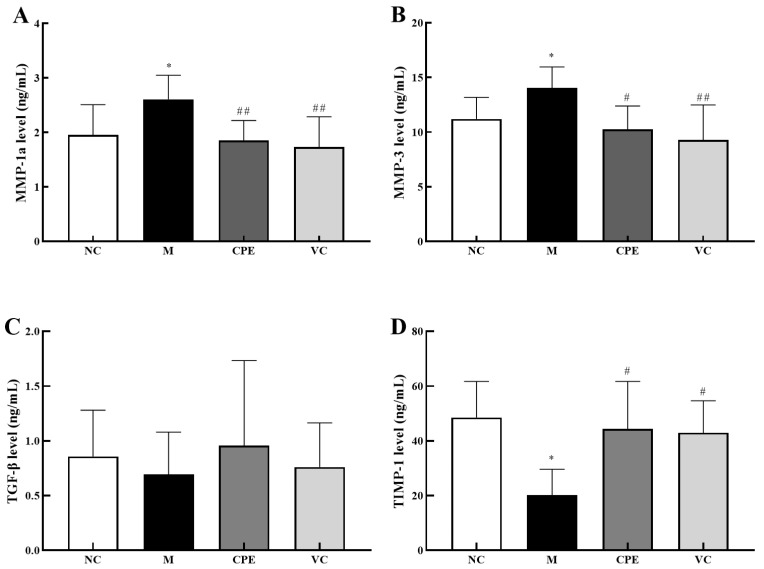
MMP-1a, MMP-3, TIMP-1 and TGF-β levels in mice skin. Note: (**A**): MMP-1a content in mouse skin tissue; (**B**): MMP-3 content in mouse skin tissue; (**C**): TGF-β content in mouse skin tissue; (**D**): TIMP-1 content in mouse skin tissue. “*” indicated that the date had statistically significant differences (*p* < 0.05) compared with group NC. “#” and “##” indicated that the date had statistically significant differences (*p* < 0.05) and extremely significant (*p* < 0.01) compared with group M, respectively.

**Table 1 nutrients-14-04659-t001:** Sequences of primers used for RT-PCR.

Gene	Forward Primers (5′-3′)	Reverse Primers (5′-3′)	Size (bp)
*MMP-1a*	ATAGATTCATGCCAGAACCTGA	TGCCTTTGAAATAGCGGACT	120
*MMP-3*	AATCAGTTCTGGGCTATACGA	TCGATCTTCTTCACGGTTGC	95
*TGF-β*	AACAATTCCTGGCGTTACCTT	CTTGGTTCAGCCACTGCCGTA	106
*TIMP-1*	TCCCAGAACCGCAGTGAAG	ACGCCAGGGAACCAAGAAG	93
*GAPDH*	GCGACTTCAACAGCAACTCCC	CACCCTGTTGCTGTAGCCGTA	122

RT-PCR, reverse-transcription polymerase chain reaction; *MMP-1a*, matrix metalloproteinase 1a; *MMP-3*, matrix metalloproteinase 3; *TGF-β*, transforming growth factor-β; *TIMP-1*, metalloproteinase 1; *GAPDH*, glyceraldehyde-3-phosphate dehydrogenase.

## Data Availability

Not applicable.

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
