# Peer review of "Antioxidant Capacity and Protective Effect of Cow Placenta Extract on D-Galactose-Induced Skin Aging in Mice"

_nutrients, 2022, doi:10.3390/nu14214659_

Round 1

Reviewer 1 Report

Dear Authors,

Introduction

1.      It is mentioned that- Although there are many studies on skin care products for external use to delay skin aging, there are few studies on oral anti-aging drugs… kindly do mention few points in favour of oral administration for the said purpose

2.      It is mentioned that- major structural molecules of the skin[6, 7]. Through glycosylation, covalent binding of sugars to proteins and subsequent generation of advanced glycation… kindly verify the flow

3.      It is mentioned that-However, the placenta of dairy cows is widely sourced, but it is not used effectively and is even discarded, causing environmental  pollution[16]. Kindly consider mentioning few features of the former prior to the above statement

Methodology

1.       It is mentioned that- farm in Sichuan Province, cleaned the placentas with saline to remove residual blood and store at -20 °C…. kindly revise

2.       Methodology needs to be narrated in past tense. kindly verify the narration too

3.       The Elisa kits were purchased- kindly revise the acronym

4.       Total RNA of skin was separated by an Animal tissue RNA extraction kit… kindly revise

Results

1.      and the average TEAC of CPE was 19.56 μmol/g … kindly expand the acronyms once prior to their use

2.      Effect of CPE on antioxidant capacity in vitro- kindly mention the inference after the  evaluation of the three different factors considered

3.      CAT, GSH, GSH-Px, POD, SOD, were highly significant higher (P < 0.001). MDA was very significantly lower (P < 0.001) in the CPE and VC groups compared with the M group. Kindly verify as the former parameters group is not mentioned

4.      Graphs should mention in the legend what does #,* represent

5.      Kindly consider mentioning the inference of the results under the respective subheadings

Discussion

1.      It is mentioned that- In addition, the body weight growth rate of mice fed with CPE was significantly higher than that of mice fed with vitamin C and healthy mice, and the final body weight of mice fed with D-gal was significantly higher than that of mice injected with vitamin C… kindly comment and verify

2.      It is mentioned that- compared to healthy mice, indicating that D-gal injection caused mice caused severe oxidative damage… kindly revise

3.      Kindly comment on the limitations of the study

4.       Kindly comment on the future directions at the end of the discussion

5.      Kindly comment whether cow placenta and other animal placenta have to be compared with similar methodology  or different preparation process of cow placenta have to be evaluated for their activity

Conclusion

1.       It is mentioned that- In this papar, we eveluted the antioxidant activity of CPE… kindly revise the terminologies

Regards

Reviewer 2 Report

Please see in the attached file

Reviewer 3 Report

The authors present a study on cow placenta extracts' effect on reducing skin aging processes in vivo and in vitro.

It seems the study is meticulously performed and well described, but I have comments on the Introductory part, where faulty claims are made, some claims are too general, and actually do not explain the background properly.

I suggest that the authors first describe what it is that placentas contain and why these may be of significance for counteracting aging and skin aging in general, why cow placentas were chosen (specific qualities or just availability) Then there should be a clear hypothesis. The authors do not provide any objective or hypothesis based on clear evidence. See comments and examples below:

The abstract should be rewritten. Many sentences do not make sense or are grammatically incorrect, therefore not readily understood. 'Cow placenta in the pasture is often discarded...' Is this universal? Or do you mean a specific place/country/geographic area? Also, I am not certain that discarding placentas can be called pollution. Organic matter is readily used up in nature and may even help promote a healthy eco-system.

Please also make sure that any abbreviations are first given in full and the abbreviation in parentheses the first time it is used. There are 11 abbreviations in the abstract that are not explained. All abbreviations in the text and figures/tables should likewise be explained even if common (T-AOC, GSH, GSH-Px, POD, SOD, CPE, CAT, MMP-1a, MMP-3, TGF, TIMO, D-gal, UV, ROS, DNA. RNA, ATP,...)

'..Risk of skin disease..' with skin ageing is very general. Several skin conditions are more prevalent in young skin (acne, autoimmune disorders, pediatric skin conditions). Wrinkling, roughness, dryness etc, is indeed more common in older skin, but can be regulated by correct skin care or is only of cosmetic importance. Skin cancer is related to sun exposure rather than skin aging per se. It is not completely correct to claim that because of wrinkling or roughness finding an oral medication is essential. I suggest the authors find other reasons for the importance of their study, for instance, that knowledge in preventing aging is in general of high importance, where finding new products and using already available sources may provide good research in this area.

You write that: 'there are few studies on oral anti-aging drugs. Therefore, it is essential to find new anti-aging oral medicines'. This is a strong claim. I wouldn't say it is essential to find new anti-aging oral medications based on the fact that there are few studies on this. And because aging skin is wrinkled and rough. If you wish to point out the cosmetic benefits you should rather have a background on the importance of feeling and looking young and how this may impact the quality of life.

When pointing out that placentas of dairy cows are not used effectively, do you mean to say that other placentas (other animals, or non-dairy cows) are not useful? Or do placentas from other mammals have the same qualities and can also be used, but dairy cow placentas are simply more easily available. It is wrong to claim that specifically dairy cow placentas cause pollution.

Several sentences are not structured correctly. E.g: Through glycosylation...while causing... and leading to...
Please read through and make sure the grammar and English is correct throughout.

Round 2

Reviewer 1 Report

The revisions executed by the authors are satisfactory

Reviewer 2 Report

The comments were addressed by the authors in the revised manuscript of nutrients-1993444.R1

Reviewer 3 Report

The authors have adequately addressed my comments.